# Patients' experiences of life after bariatric surgery and follow-up care: a qualitative study

Karen D Coulman ,[1] Fiona MacKichan ,[1] Jane M Blazeby,[1,2]
Jenny L Donovan,[1,3] Amanda Owen-Smith[1]

[1]Population Health Sciences, Bristol Medical School, University of Bristol, Bristol, UK
[2]Division of Surgery, Head and Neck, University Hospitals Bristol NHS Foundation Trust, Bristol, UK
[3]NIHR CLAHRC West, University Hospitals Bristol NHS Foundation Trust, Bristol, UK

**Correspondence to**
Dr Karen D Coulman;
Karen.Coulman@bristol.ac.uk

## ABSTRACT

**Objectives** Bariatric surgery is the most clinically effective treatment for people with severe and complex obesity, however, the psychosocial outcomes are less clear. Follow-up care after bariatric surgery is known to be important, but limited guidance exists on what this should entail, particularly related to psychological and social well-being. Patients' perspectives are valuable to inform the design of follow-up care. This study investigated patients' experiences of life after bariatric surgery including important aspects of follow-up care, in the long term.

**Design** A qualitative study using semistructured individual interviews. A constant comparative approach was used to code data and identify themes and overarching concepts.

**Setting** Bariatric surgery units of two publicly funded hospitals in the South of England.

**Participants** Seventeen adults (10 women) who underwent a primary operation for obesity (mean time since surgery 3.11 years, range 4 months to 9 years), including Roux-en-Y gastric bypass, adjustable gastric band and sleeve gastrectomy, agreed to participate in the interviews.

**Results** Experiences of adapting to life following surgery were characterised by the concepts of 'normality' and 'ambivalence', while experiences of 'abandonment' and 'isolation' dominated participants' experiences of follow-up care. Patients highlighted the need for more flexible, longer-term follow-up care that addresses social and psychological difficulties postsurgery and integrates peer support.

**Conclusions** This research highlights unmet patient need for more accessible and holistic follow-up care that addresses the long-term multidimensional impact of bariatric surgery. Future research should investigate effective and acceptable follow-up care packages for patients undergoing bariatric surgery.

## INTRODUCTION

Over 650 million or 13% of adults worldwide suffer from obesity (body mass index (BMI) ≥30 kg/m²), representing a tripling of figures since 1975.[1] Obesity is associated with an increased risk of type 2 diabetes, cardiovascular disease, certain cancers and premature death.[2,3] Within this population, people with severe and complex obesity (BMI ≥40 kg/m², or 35–40 kg/m² with another

### Strengths and limitations of this study

► Patients who had undergone all three main types of bariatric procedures across two UK centres were included in the research.
► A detailed qualitative approach was used, allowing participants to relate their own experiences in terms that were relevant for them.
► A rigorous approach to analysis was undertaken, including independent coding of initial transcripts by three researchers, and agreement of emergent themes throughout analysis with at least one other qualitative researcher.
► It is not known whether similar themes would be found with participants in other centres.
► Findings relating to follow-up care may be less generalisable to healthcare systems with different service pathways and funding structures.

significant health problem that could be improved by weight loss) suffer the greatest health burdens and are at the highest risk of premature death.[4,5] In addition to the physical and metabolic health burdens, people with severe and complex obesity are more likely to suffer with psychological disorders such as depression, anxiety and disordered eating, and reduced health-related quality of life (HRQL).[6,7] These individuals also suffer from social stigma and discrimination related to their weight,[6,8] which is in turn associated with adverse physical and psychological outcomes.[8,9] Thus, any interventions to treat severe and complex obesity should consider the impact on these psychosocial outcomes in addition to traditional clinical and metabolic outcomes.[10,11]

Bariatric surgery, combined with behavioural change and dietary management, is the most clinically effective treatment for people with severe and complex obesity, in terms of weight loss and the improvement of comorbidities such as type 2 diabetes.[5,12,13] The three main types of bariatric operations performed in the UK include the Roux-en-Y

gastric bypass (RYGB, 53.9% in 2011–13), the sleeve gastrectomy (SG, 21.4%) and the adjustable gastric band (AGB, 21.4%).[14] More recent international data indicate that the SG (46.0%) and RYGB (38.2%) are the most common bariatric operations worldwide with AGB decreasing in recent years (5.0%), and the one-anastomosis gastric bypass now gaining popularity.[15] Each of these procedures works slightly differently; mechanisms include restriction in the amount of food able to be consumed, reduction in hunger, improvement in satiety, shift in food preferences, as well as altered gut hormones, bile acids and vagal signalling.[16] While there are lots of non-randomised studies in this field, there are very few well designed and conducted randomised controlled trials with long-term follow-up. This means that true comparative assessments of RYGB, SG and AGB are absent from the literature. A current UK study has recently completed recruitment (n=1351), with the primary end point at 3 years. This will be the first pragmatic large-scale study examining all three procedures.[17]

Studies which have examined HRQL after each procedure are often poorly conducted with few including baseline data and comprehensive assessments of HRQL. Some show certain aspects of HRQL to improve but not others.[11 12 18] Previous qualitative research has highlighted the complex and changeable nature of the psychosocial impact of bariatric surgery, helping to shed light on some of these inconsistencies in the HRQL literature and emphasising the importance of long-term postoperative support in helping patients manage these changes.[10 19] Previous research has also reported attendance at follow-up visits to be associated with better weight loss outcomes after bariatric surgery.[20] Follow-up care is thus important to optimise clinical and psychosocial outcomes of bariatric surgery. However, bariatric surgery follow-up care has been reported to vary greatly across the UK,[21] and current UK and US bariatric surgery guidelines focus on surgical and metabolic outcomes, with limited guidance on how to support psychological, social and lifestyle changes that affect patients' HRQL.[5 22] Nevertheless, previous work has highlighted the importance of these multifaceted aspects of HRQL to patients who have undergone bariatric surgery and recommendations are needed on how best to support patients after surgery to optimise these outcomes.[23]

In seeking to evaluate and provide recommendations on bariatric surgery follow-up care, the patient's perspective can provide valuable information.[24] Qualitative research is useful to explore patients' perspectives as it seeks to gain the insider's view on how people view, experience and make sense of their social world.[25–27] The primary focus of most previous qualitative research in bariatric surgery has been on patient experiences of outcomes of surgery rather than experiences of follow-up care.[10 19] Studies that have reported on aspects of care have identified patient need for longer follow-up after bariatric surgery, better access to psychological support and the ability to communicate with health professionals between routine appointments.[19 28–36] However, most of these studies were single centre[29–32 34 36] or reported findings from select groups, such as patients that had undergone one type of bariatric procedure only (eg, AGB)[29 30 32–35] or had experienced negative outcomes such as weight regain or substance abuse issues.[28 29 32 34] A recent systematic review by Parretti *et al* identified few studies focusing on patients' experiences of follow-up care after bariatric surgery in the longer term, and recommended that primary studies in this area were needed.[19] The objectives of this study were to: (1) Investigate experiences of life after bariatric surgery including follow-up care in the long-term across people who had undergone all three main types of UK bariatric procedures and (2) Use these findings to provide recommendations for follow-up care.

## METHODS

Patients who had undergone a primary operation for obesity at two publicly funded bariatric surgery centres in the South of England were eligible to participate in the research. Patients were identified by health professionals at each hospital using databases and clinic lists and sent information about the research. Interested patients contacted the researcher directly (KDC). For initial interviews, patients were sampled purposively, aiming for maximum variation in gender, age, starting BMI, type of operation and time since operation. Emerging findings from analysis of initial interviews guided sampling for remaining interviews.[37] Sampling continued until themes were well established with few or no new insights gained from additional data collection.[26 37] This study was undertaken as part of a wider study to develop a core outcome set for bariatric surgery (see online supplementary document S1 for protocol).[23]

Interviews were chosen as the method of data collection for this study due to the sensitive and complex nature of living with bariatric surgery, and to allow individual participants' experiences to be explored in detail. Interviews were semistructured to provide some consistency in topics discussed between interviews, while allowing flexibility to adapt each interview to the participant. Thirteen participants were interviewed in their homes, four in a private research room at one of the two participating hospitals, one in a private room at the University and one over the telephone at their request. Interviews lasted between 44 and 110 min.

Written informed consent was taken and interviews conducted according to an outline topic guide, which evolved iteratively as the research progressed (see online supplementary document S2 for final version). Findings reported in this paper mainly relate to the sections of the topic guide 'Actual outcomes of surgery' and 'Actual experiences of follow-up care'. Relevant demographic and clinical information were also collected (online supplementary document S3). All interviews were conducted and audio recorded between February 2013 and November 2014, by a female researcher (KDC) who

was a PhD student and registered dietitian. KDC underwent training in qualitative research methods and was supervised by two experienced qualitative researchers (AO-S and FM). An initial telephone conversation was held with each participant to discuss the study and arrange the interview. Participants were otherwise not previously known to the researcher prior to interview. The researcher introduced herself as a PhD student to participants. She did not reveal her professional background as a registered dietitian unprompted but did not seek to hide it if participants asked. Field notes, which provided important contextual information to aid data analysis, were made as soon as possible after each interview.[38]

Recorded interviews were transcribed verbatim, and transcriptions checked for accuracy by KDC. Thematic analysis was undertaken, using techniques of constant comparison to code data and identify emerging themes.[37 39] As the aim of the study was to broadly investigate patients' experiences of surgery, including outcomes and aspects of care, this inductive approach to analysis was chosen to ensure that themes developed were strongly linked to the data. Coding was completed for all transcripts by KDC, with a sample of transcripts independently coded by two other experienced qualitative researchers (AO-S and JLD) (see online supplementary document S4 for final coding framework). Differences in interpretation were resolved through discussion. Initial codes were built into coding structures and themes were identified. Coding and data management were facilitated using NVivo 10 software.[40] Detailed descriptive accounts were written by KDC for each small batch of interviews, which described data relating to each theme and its constituent codes. It was at this stage that relationships between themes were identified, leading to the development of higher-order categories which encompassed inter-related themes. The coding and descriptive account were completed for each batch of interviews prior to recruiting additional patients so that emerging themes could be followed up to enrich subsequent interviews. Finally, large matrices were created to compare themes and categories across all participants and summary descriptive accounts were written wherein the concepts overarching all themes and categories crystallised.[39] AO-S, FM, JLD and JMB reviewed all descriptive accounts and made suggestions about further links between themes, categories and concepts.

### Patient and public involvement

The idea for this research was based on the lead author's experience of working with patients over several years in a bariatric surgery service, as well as discussion with a representative from a relevant patient charity. This patient representative reviewed and provided feedback on the research proposal submitted for funding. After the study received funding, two patients who had undergone National Health Service-funded bariatric surgery were recruited as patient research partners and reviewed and provided feedback on the interview topic guide, and all written patient information (including study recruitment documents, and the final study summary disseminated to participants).

## RESULTS

Of 48 patients invited, 17 agreed to take part in interviews (mean time since surgery 3.11 years, range 4 months to 9 years), although two others (spouses of existing participants) were opportunistically recruited as the research was ongoing. Twelve of the 19 participants were female, and the mean age was 51.1 years. All reported their ethnicity to be 'White British', and 17 had already undergone surgery (table 1). The analysis presented draws on interview data from the 17 participants that had undergone surgery.

Bariatric surgery was a life-changing journey for participants, impacting on several different areas of their lives. The overarching concepts of 'normality' and 'ambivalence' emerged from analysis of data on patients' experiences of adapting to life after surgery (figure 1). Analysis of data relating to experiences of follow-up care was conducted separately and characterised by two concepts—'abandonment' and 'isolation' (figure 2). Results are presented according to overarching concept with participant quotes used to support the description of each concept.

### Adapting to life after surgery: normality and ambivalence

Throughout several areas of their lives, participants were striving to be more 'normal' after bariatric surgery. This related to different aspects of their lives categorised as physical health, psychological health, eating patterns and hunger, body image, weight and social functioning (figure 1). Participants experienced many positive changes that undeniably brought them closer to their idea of normality. However, participants also described things that did not change, for which they still felt abnormal. Some also experienced changes perceived as negative or difficult to deal with, which made them feel more abnormal and required a process of adjustment. This was acknowledged as a 'trade-off' or the 'price to pay' (P08) for the benefits gained. The complexity of the changes experienced highlighted the ambivalence of living with the results of bariatric surgery. Despite the challenges, all participants felt the surgery was a good decision: 'I don't regret it for a minute. Despite all the complications and issues' (P14).

### Normality

All participants reported an improvement in activity and mobility levels and/or their ability to carry out 'normal' activities of daily living following surgery: 'I'm more mobile, I can tie my shoelaces, shower properly…my life has changed for the better' (P10). Participants also reported several positive changes related to physical and psychological health including a reduction in medications required (eg, for diabetes), an improvement in physical symptoms (such as joint pain), self-confidence

**Table 1** Characteristics of participants

| Participant | Gender | Age range (years) | Marital status | Employment status | Type of surgery | Time since surgery (years) |
|---|---|---|---|---|---|---|
| P01 | Female | 60–70 | Married | Retired | RYGB | >5 |
| P02 | Female | 50–60 | Married | Unemployed | RYGB | <1 |
| P03 | Female | 30–40 | Married | Employed* | RYGB | 1–2 |
| P04 | Female | 60–70 | Married | Retired | AGB | >5 |
| P05 | Male | 40–50 | Married | Employed | RYGB | <1 |
| P06 | Female | 30–40 | Married | Employed | Awaiting surgery | N/A |
| P07 | Female | 40–50 | Married | Employed | RYGB | >5 |
| P08 | Male | 60–70 | Married | Employed | AGB | >5 |
| P09 | Female | 40–50 | Married | Unemployed | SG | 1–2 |
| P10 | Male | 30–40 | Co-habiting | Self-employed | SG | 2–5 |
| P11 | Female | 40–50 | Married | Employed | SG | <1 |
| P12 | Female | 50–60 | Married | Self-employed | SG | 1–2 |
| P13 | Male | 50–60 | Widowed | Employed | RYGB | <1 |
| P14 | Female | 40–50 | Married | Employed | AGB and RYGB | >5 |
| P15 | Male | 60–70 | Married | Retired | RYGB | 1–2 |
| P16 | Female | 60–70 | Married | Retired | Awaiting surgery | N/A |
| P17 | Male | 40–50 | Married | Employed | AGB | 2–5 |
| P18 | Male | 50–60 | Co-habiting | Employed | AGB | 1–2 |
| P19 | Female | 30–40 | Separated | Employed | AGB | 1–2 |

*'Employed' status includes those employed both full time and part time.
AGB, Adjustable gastric band; N/A, not applicable; RYGB, Roux-en-Y gastric bypass; SG, Sleeve gastrectomy.

and psychological well-being: 'I feel healthier mentally in my head, like I want to get out there' (P09).

Some participants described an improved or more 'normal' relationship with food after surgery, whereby they had retrained their mind to focus on 'eating more sensibly' rather than thinking they were 'on a diet' (P11). Others experienced no real change to their relationship with food, feeling as though they still had to be 'on a permanent diet' (P19), or continued to use food as way of coping with difficult emotions which remained: 'I still have an awkward relationship with food…still have the same demons…I probably rely on food to deal with certain emotions' (P14).

All 17 of the participants had lost a large amount of weight since having surgery, however, eight had regained some of this weight. Participants reported feeling

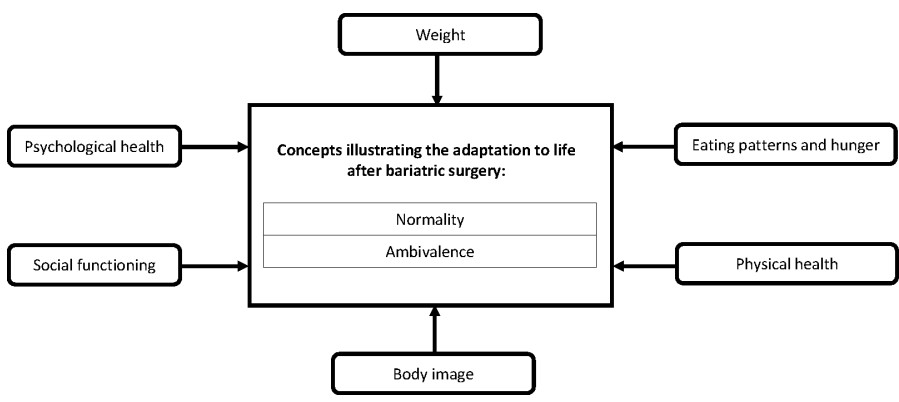

**Figure 1** Concepts and categories illustrating the adaptation to life after bariatric surgery including an example of supporting themes for one category.

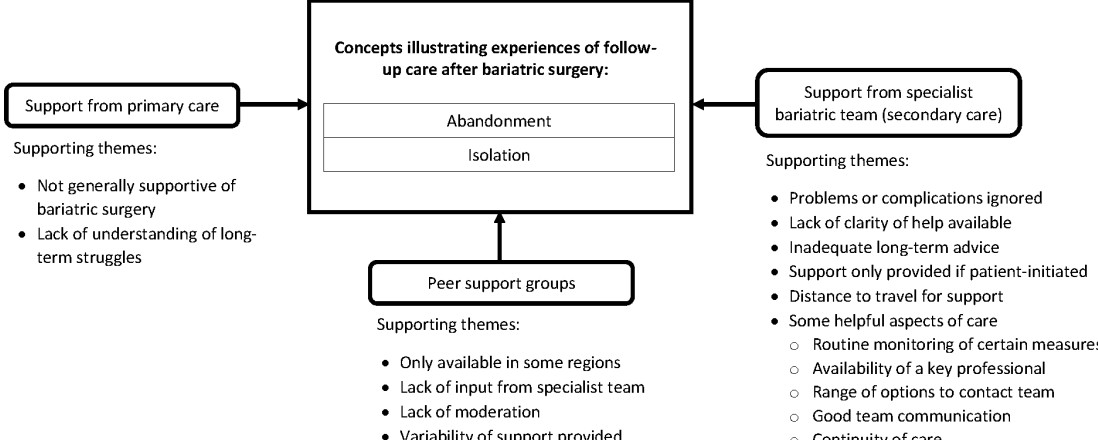

**Figure 2** Concepts and categories illustrating the experiences of follow-up care after bariatric surgery including supporting themes.

distressed by this as they did not want to return to the way they were: 'That was a real horrendous thing for me to see my weight go up a bit after all I'd gone through to get it down…' (P07). However, a couple described being reassured by health professionals that it was normal to experience some weight regain. The majority related their weight regain to a gradual increase in appetite and/or portion sizes over time (which had initially decreased after surgery), and a feeling that the surgery was not as effective as it had been: 'I don't seem to be getting the urge to stop quicker, like I did before' (P18).

The majority of participants reported developing loose-hanging excess skin following their massive weight loss, which challenged their sense of normality. Although they were pleased to be a more 'normal' size, some felt ashamed of how abnormal their body looked without clothes on. Skin removal surgery was a costly option, so some had learnt to live with the excess skin; however, a few found the excess skin to be particularly problematic, impacting on their mental health and relationships: 'My husband doesn't like the excess skin…and that's one of the reasons why I must do something about it, because…I know I look like a bag of s**t' (P12).

### Ambivalence

Although improvements to existing health problems were important benefits of the surgery, five participants reported developing new health problems postsurgery, including micronutrient deficiencies, menstrual problems, brittle bones, low blood pressure and cardiac issues: '…you give up one set of health implications but you get given another set in its place…' (P07). Some participants still suffered with several food intolerances and/or frequent gastrointestinal symptoms many years after surgery, which they reported resulted in a poorly balanced diet: 'I can't eat bread or meat…That's one of the small prices I have to pay…my intake of food is nowhere near balanced…' (P08).

Difficulties were described in developing new coping strategies to replace food, which had previously been a 'comfort blanket': '…all your insides are different but your brain…no different whatsoever…that for me was the hardest thing to adjust to, because my brain was still telling my stomach I was hungry but obviously I couldn't [eat]…' (P03). One patient described developing an alcohol dependency postsurgery (which they had eventually overcome), and two participants mentioned the need for more psychological input to help with their adjustment following surgery: 'There was no formal counselling…and that might be a good idea to find out why we eat so much, why are we addicted to food…' (P04).

Ambivalence was also evident in participants' experiences of social functioning and stigma. Participants reported receiving positive attention due to their weight loss: '…people tell you 'you look brilliant'…that is the good side of it' (P17). For some, however, this led to mixed emotions at the revelation of 'how negative people saw you before' (P07). Others described receiving less negative attention and feeling less socially stigmatised due to their obesity: 'I can walk down the road now and not get such the bad looks as I used to.' (P04). However, a number of participants had experienced a new type of social stigma at having taken the 'easy way out' (P02) by having surgery (eg, not achieved weight loss through the 'normal' means). Some were ashamed to tell others they had undergone surgery for fear of this reaction.

Social and family eating situations could also cause anxiety for some due to attracting attention for only eating very small amounts, or unpleasant and embarrassing gastrointestinal symptoms which could arise when eating. For some this had remained an issue several years following surgery causing disruption to relationships: 'It disrupts life because I can be eating and whether it's the wrong food, a mouthful too much…I've got to go out and she can hear me retching, and it puts her off her food' (P08). Others were able to adapt or reported their social life had 'come back' (P10) gradually as food tolerance improved.

**Table 2** Participant quotes to support positive experiences of follow-up care

| Positive aspects of care | Quotes |
| --- | --- |
| Routine monitoring of certain measures | 'It was good having my bloods done so I could check what my levels were like, that was quite useful for me…routine monitoring was good.' (P07) |
| The availability of a key health professional; ability to contact the team using a range of contact options | 'If I couldn't get hold of her (dietitian) straight away on the phone I'd send an email and it would either be answered the same day or the next day.' (P09) |
| Good communication between team members | 'It's quite a tight little team….you might not necessarily speak to the best person, but they will come together in their meeting and you'll get the best outcome.' (P19) |
| Continuity of care | 'You didn't see twenty different people. It was 'the team'…the same faces…I like that. I don't want to see somebody who's different don't know you…' (P08) |
| Overall positive view of care | 'The follow-up care I've had has just been 110%, if I've had a problem I would ring and…I would get an appointment…Someone has always been there for me…' (P01) |

## Experiences of follow-up care: abandonment and isolation

Participants explained that follow-up care received after surgery was mainly provided by the specialist bariatric surgery team (although what this entailed was highly variable), with little support from their general practitioners (GPs). Only a few participants described feeling well supported overall, and all of these had undergone their surgery less than 2 years previously. However, most described at least one aspect of follow-up care which they found helpful. These included: (1) the routine monitoring of certain measures (eg, weight, nutritional blood tests); (2) the availability of one key health professional (generally a specialist dietitian or nurse), who was easy to contact on an ad hoc basis; (3) the ability to contact the bariatric team using a range of contact options (eg, telephone, email); (4) good communication between team members and (5) continuity of care (eg, being able to see the same professionals at every appointment) (table 2).

Overall, however, there was a sense of abandonment and isolation in participants' accounts of follow-up care. This related to their experiences of postoperative support from the specialist team, primary care professionals and peer support groups (figure 2). Participants felt that health professionals did not always appreciate the long-term implications of life after surgery, or even if they did, services were not set up to support them adequately: 'It happened eight years ago so no one thinks you may have any hang-ups, issues, concerns about it…the implications of the changes it makes people don't really appreciate, it's an old record, old news' (P07).

### Abandonment

Some participants felt that problems or complications they experienced following surgery were ignored or not dealt with properly, or there was a lack of clarity of who to go to if they experienced problems. P07, for example, felt her postoperative problems were dismissed by the specialist team, and that she 'was upsetting someone's figures by having complications'. P12 experienced a problem with one of her surgical wounds which wouldn't heal and wasn't sure who to go to about it. She felt 'quite abandoned' and dealt with it mainly on her own. Abandonment also related to the feeling they had been given inadequate information or guidance about life following surgery: 'They give you loads of information about what to do in the first six weeks and then there's nothing…' (P04).

Abandonment was also evident in accounts of support only being provided when patients themselves initiated contact: 'I feel that as long as you didn't contact them then you will be left alone…' (P15). Concerns were raised for others whom they perceived less likely to seek help proactively: '…these people aren't coming forward to explain that they're having problems because they don't want to feel like a failure…' (P09). P18 expressed disappointment that he had not been sent any appointments post-operatively and felt he had been left 'in limbo' to 'get on with it' himself. He had not asked for help and was under the impression that it would only be appropriate to contact the team if you were having complications: '…obviously if I was in excruciating pain from the operation I suppose, I could have gone back…' (P18).

Most participants also reported feeling abandoned by their GPs who were not usually supportive of them having undergone bariatric surgery and did not 'fully appreciate the struggles that you have' (P14) in the long-term. However, a minority of participants described feeling well-supported by their GPs who recognised the long-term health benefits of bariatric surgery: '…with being my dad's doctor, he sees that hopefully I won't have the same problems…he's done everything he can to help me…' (P05).

### Isolation

Several participants did not live locally to the hospital where the specialist team were located. This presented a barrier to accessing follow-up care, which some felt could contribute to feelings of isolation: 'From this side of the county it's (hospital) extremely difficult to get to…I can understand an awful lot of people thinking 'if I ring [hospital] they're going to say come over and see me and that is so difficult to get to…I won't bother' (P15).

Equally participants described how local primary care services were unable to support them compounding their feelings of isolation: 'Unless they've (GP surgery) read my notes they don't even know I've got one (a gastric band)' (P04), and 'They (GP surgery) were very much like 'it's a secondary care issue, it's not primary care'' (P07).

Isolation was also apparent in participants' experiences of bariatric surgery peer support groups. Although not part of medical care, these represented an important source of support. These groups were typically run by patient volunteers, with limited or no input from health professionals. Some participants had access to these groups in their local areas, whereas others did not. Those unable to access a group felt this contributed to their sense of isolation postsurgery: '…there's meetings where you can meet other people who've had the [gastric] band…but there's no local ones for me…if people said, 'If you do eat it, it's going to hurt but it will go, and this is the reason it's hurting,' then I could have dealt with it a little bit better.' (P17). Those that had accessed these groups described variable experiences. Some found them supportive, for example, P01 who continued to attend several years postsurgery, whereas others had negative experiences and felt quite isolated from other members. P19, for example, had disengaged from her local group which she described as being very 'cliquey' with members using the group mainly to emphasise negative experiences or 'how to cheat the band'. Many felt that peer support groups including 'a chairman' (P15) knowledgeable in the results of bariatric surgery should be part of routine clinical care to improve accessibility of peer support and ensure consistency of information discussed.

## DISCUSSION

This qualitative study found that bariatric surgery impacted participants' physical and psychological health, eating behaviours, weight and social functioning. The overarching concepts of normality and ambivalence illustrated their lived experience following bariatric surgery. Normality was evidenced through participants' relief at feeling more normal in some ways (eg, improved ability to undertake daily activities), yet feeling less normal in other areas, including the development of excess skin and difficulties eating 'normally' in social situations. Although participants experienced many positive health changes, they also experienced changes which were negative or difficult to adapt to, such as an inability to rely on emotional eating as an entrenched coping mechanism, perceived bodily deformity as a result of excess skin and the destabilisation of important relationships. These complexities highlight the ambivalence of living with the outcomes of bariatric surgery. In coping with these changes, participants received varying levels of care from specialist health professionals and GPs. Although there were some positive experiences, 'abandonment' and 'isolation' characterised most follow-up care experiences. This included feeling unsupported with postsurgery

problems (other than serious complications), lack of guidance with long-term lifestyle changes, lack of understanding from GPs and limited peer support. However, all participants felt that undergoing the surgery was a good decision despite the difficulties. These findings are important in helping to define future follow-up care packages to better address the complex changes experienced after bariatric surgery.

Our findings are consistent with previous qualitative research on patient experiences of living with outcomes of bariatric surgery which depicted the complexities on patients' sense of normality and the 'give and take' or ambivalent nature of the changes experienced.[10 41–43] This study strengthens the evidence for the individual and nuanced nature of how bariatric surgery changes people's relationship with food in different ways, and changes over time, indicating the need for individualised dietary and psychological support at different time points.[10 28 41 43 44] The importance placed by participants on the social impact of bariatric surgery was also noted in a recent UK study by Graham *et al*.[45] These issues, including difficulties with social and family eating, should be given more attention in follow-up care. Our study confirms previous qualitative findings on the importance of continuity of care,[19] the ability to access professional advice (often from the specialist dietitian) between appointments via telephone or email,[31] the lack of psychological support after surgery[19 28–30 32 33 36 46] and the need for moderation in patient support groups.[33 34] Previous studies have related patients' views that GPs were not equipped to adequately support them postsurgery.[19 30 31 47] This was also evident in our study with most participants describing negative experiences with GPs in relation to bariatric surgery, and feeling they were unable to offer adequate support. Despite this, several participants would have preferred to access support locally due to living remotely.

Our study expanded findings on patient experiences of bariatric surgery follow-up care as being characterised by feelings of abandonment and isolation, with views that services were not set up to support long-term issues. Abandonment was also evident in a study by Jumbe and Meyrick who described a 'postsurgical cliff' with patients receiving intensive support prior to bariatric surgery and then feeling abandoned after surgery.[36] Similar to our study, they described how postoperative support was reliant on patient-initiated contact. Previous research with people living with obesity suggests they may delay or avoid seeking healthcare due to societal and medical stigmas.[48 49] This has also been reported by Throsby who conducted a UK-based ethnographic study within a surgical weight management clinic.[50] She described examples of patients struggling with their eating habits and weight postsurgery, and the shame they felt at doing 'badly' after undergoing publicly funded surgery. The author argued that this 'moral weight' could lead to patients not seeking help when most needed.[50] Similarly, feelings of shame and failure at not having met the perceived postoperative

expectations was one reason cited by Australian patients for non-attendance in bariatric surgery aftercare.[30]

The main strength of this research is that a detailed qualitative approach to data collection was used, whereby participants were given the time and flexibility to relate their own experiences in terms that were relevant for them. A rigorous approach to analysis was undertaken, including independent coding of initial transcripts by three researchers, and discussion and agreement of emergent themes throughout analysis with at least one other qualitative researcher. A limitation of this study is the lack of ethnic diversity represented within the sample. Low numbers of people from ethnic minority groups undergo bariatric surgery in the UK (1303 between 2011 and 2013, 7.7% of total procedures), making it difficult to identify eligible people for qualitative studies.[14] A strength of this research is that we were able to over-represent male participants within our sample (41% of the 17 postoperative participants compared with 24% who undergo bariatric surgery nationally), which has been a limitation of previous qualitative studies in this area.[14 28 30 32 34–36] An additional strength was the inclusion of a clinically diverse group of patients who had undergone all three main types of bariatric procedures in the UK and who were at a broad range of time points postsurgery. Participants were also recruited from two UK centres with different follow-up programmes and health professional teams. It is not known, however, whether similar themes would be found with participants in other centres. The findings relating to follow-up care may be less generalisable to healthcare systems with different service pathways and funding structures.

Taken together with previous literature, our findings highlight that current bariatric surgery follow-up care provision is not often aligned with patient need. Patients highlighted the need for a flexible and long-term approach to follow-up care from a multidisciplinary health professional team. This includes both routine and open appointments, moderated peer support groups and different methods of contact (eg, telephone, online in addition to face to face). These recommendations are also in accordance with the recently published 2019 UK psychological guidelines for bariatric surgery which recommend a flexible and individualised approach to postoperative psychological support, including routine screening at 6–9 months postsurgery to identify support needs.[51] In addition to individual dietary and psychological support, services should consider how to better support patients in developing strategies to cope with family and social difficulties post-surgery. This may include actively engaging family and close friends in preoperative preparation and/or postoperative interventions. Future research is needed to define and evaluate an effective and acceptable follow-up care package that could be consistently applied across bariatric surgery centres. This may include the optimal systems or pathways to identify and support those who need the most help but are the least likely to seek it, ways of engaging family and social support and delivering moderated peer support groups. The relative merits of delivering follow-up care in specialist or community-based health services or how it might be shared between the two should also be investigated.

**Acknowledgements** The authors would like to thank the patient research partners who advised on this research for their valuable input, as well as the study participants who gave up their time to take part in the research and the health professionals at the centres who helped with study recruitment.

**Contributors** KDC led the study design, data collection and data analysis as part of her PhD research, and drafted this manuscript. AO-S, FM and JMB were KDC's PhD supervisors and advised on study design, data collection and analysis, and provided comments on this manuscript. JLD advised and contributed to data analysis and provided comments on this manuscript. All authors approved the final submitted manuscript.

**Funding** KDC was funded by a National Institute for Health Research (NIHR) Doctoral Research Fellowship for this research project. This work was also supported by the Medical Research Council (MRC) ConDuCT-II Hub (Collaboration and innovation for Difficult and Complex randomised controlled Trials In Invasive procedures—MR/K025643/1). This publication presents independent research funded by the NIHR and the MRC. JMB is an NIHR senior investigator.

**Competing interests** None declared.

**Patient consent for publication** Not required.

**Ethics approval** Ethical approval for the study was obtained from Northwest - Preston Research Ethics Committee (Ref 12/NW/0844).

**Provenance and peer review** Not commissioned; externally peer reviewed.

**Data availability statement** Anonymised participant data can be made available on reasonable request to the corresponding author at karen.coulman@bristol.ac.uk.

**ORCID iDs**
Karen D Coulman http://orcid.org/0000-0003-0510-4290
Fiona MacKichan http://orcid.org/0000-0002-5911-8172

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
