## [Reviewer comments · BMJ Open]

ARTICLE DETAILS

TITLE (PROVISIONAL)	Patients' experiences of life after bariatric surgery and follow-up care: A qualitative study.
AUTHORS	Coulman , Karen; MacKichan, Fiona; Blazeby, Jane; Donovan, Jenny; Owen-Smith, Amanda

VERSION 1 - REVIEW

REVIEWER	Mikaela Willmer University of Gävle, Sweden
REVIEW RETURNED	07-Nov-2019

GENERAL COMMENTS	My thanks to the Editor for the opportunity to read and review this interesting paper. I feel that it may certainly be of interest to the readers of BMJ Open. However, I would like to make some suggestions and comments that could help to further improve the paper and its impact. Introduction: I would like to see a stronger justification for this study – as you write, there have now been a fair few qualitative studies exploring experiences after bariatric surgery. You write that you have a more diverse sample of participants than other studies, but I disagree. They are all white, all come from the same part of the same country, the great majority are women and the great majority are married or co-habiting. In fact, I would say that they are a very homogenous sample! It would be interesting to see some (brief) statistics of the kind of bariatric procedures that are in use in the UK as this may differ significantly between countries. For example, gastric bands are no longer in use in Sweden, due to the poor long-term results. Additionally, as BMJ Open is not a journal focused on bariatric surgery or even obesity, I think its readership would appreciate a brief explanation of the nature and expected outcomes of the different bariatric procedures. Methods: In the Discussion section, you write that the main strength of the study lies in its “detailed qualitative approach”, but the Methods section gives very little information to justify this claim. The interviews are hardly described at all and I would like to see some more information about them – how long were they generally? If they were “mostly” conducted in the participants' homes, where
---

	were the rest conducted, and how many were conducted outside the homes? Document S2 seems to contain a large number of questions irrelevant to the aim of the study under review, and it would be very helpful to have a short description of the questions you use for your results in the Methods section. In a similar vein, I would appreciate a more detailed description of the analytical process. Results: The results, though interesting, seem a little unfinished to me, like they would benefit from another round or two of consideration. In the first, introductory paragraph (page 7, lines 11-22) of the Results section, you mention “Adapting to life after surgery” and “Experiences of follow-up care” but not the middle category (“Social functioning and stigma”). Also, since you use your two overarching concepts of “Normality” and “Ambivalence” as sub-headings in the other two categories, it seems natural to also have them in the category about social functioning and stigma? I also feel that the text under the different sub-headings isn’t always entirely relevant to the that particular theme – for example, the section starting with “The majority...” (page 9, lines 5-8) doesn’t seem to have anything to do with “Normality”. I think the entire Results section could be re-structured so that it is tighter and the categories mutually exclusive, with all text under each category relevant to that particular category and no other. I notice that you refer to Normality and Ambivalence as “concepts” – would “themes” be a better word, considering your choice of analysis method? Discussion: I note that you are quite generous with self-citation (four separate articles). There is nothing inherently wrong with citing one’s own work, of course, but this may be a bit much. Could you replace at least one or two, do you think? In exactly which way were the participants given “time and flexibility” (page 16, line 4)? Again, there is not enough detail in the Methods section to substantiate this claim. I would also like to see a discussion on the relative homogeneity of the sample in the limitations section.
--	---

REVIEWER	Sandra Jumbe Queen Mary University of London, United Kingdom
REVIEW RETURNED	12-Nov-2019

GENERAL COMMENTS	Very well written paper and adds to much needed evidence on the psychosocial impact of bariatric surgery, particularly long term A few minor comments - The introduction sets out the issues and study objective clearly. However, where you consider existing literature, there are a few
--

	other published qualitative papers similar to yours with patients who had several types of bariatric procedures that you could reflect on further. - Very clear methodology. A bit more detail about why you chose interviews as a means of data collection and thematic analysis as an approach for data analysis in this context would be good - Results - a lot of similarities with previous qualitative studies in terms of themes and subthemes which further strengthens evidence regarding life after surgery for those who have had bariatric surgery. Just one or two places where you could make your sentences shorter for better flow / easier read e.g. p10 lines 10-13 - Discussion p14 line 2 remove 'in-depth' unless you are suggesting other qualitative studies are not in-depth? p14 line 4 I am not sure 'represented' is the right word... perhaps illustrate, but just personal view p14 line 11 again I don't think 'ambivalence' is the right word here. I see that you are perhaps trying to link with the labels you have ascribed to your themes, but take a step back and think about it from a perspective away from this. I think this reflects the challenges or complexities of living with the surgical outcomes on a health perspective/ clinical level. p14 line 22 - 25 some of the additional insights you state are not new. They have been found in previous qualitative studies therefore it would be good to reflect on these. A core text in this area is a book by Meana & Ricciardi (2008). I have attached another paper which has other qualitative studies in the reference list that you could review/consider in light of your findings Are you aware of the new guidelines for psychological support that have just been published? Perhaps try to review/situate some of your discussion in context of these. The reviewer provided a marked copy with additional comments. Please contact the publisher for full details.
--	---

VERSION 1 – AUTHOR RESPONSE

Comments from reviewers:

Reviewer 1 (Mikaela Willmer, University of Gävle, Sweden):

My thanks to the Editor for the opportunity to read and review this interesting paper. I feel that it may certainly be of interest to the readers of BMJ Open. However, I would like to make some suggestions and comments that could help to further improve the paper and its impact.

We thank the reviewer for the detailed comments to improve the paper and have addressed each point individually.

Introduction:

I would like to see a stronger justification for this study – as you write, there have now been a fair few qualitative studies exploring experiences after bariatric surgery.

We have edited the final paragraph in the Introduction to make the justification for the study stronger, including the addition of the following sentence:

“A recent systematic review by Parretti et al identified few studies focusing on patients’ experiences of follow-up care after bariatric surgery in the longer-term, and recommended that primary studies in this area were needed.¹⁹”

We have edited the study objectives to improve clarity and the link with existing literature:

“The objectives of this study were to: 1) Investigate experiences of life after bariatric surgery including follow-up care in the long-term across people that had undergone all three main types of UK bariatric procedures, and 2) Use these findings to provide recommendations for follow-up care.”

You write that you have a more diverse sample of participants than other studies, but I disagree. They are all white, all come from the same part of the same country, the great majority are women and the great majority are married or co-habiting. In fact, I would say that they are a very homogenous sample!

The latest UK National Bariatric Surgery Registry report (2014) reports that only 1303 of 16,956 (7.7%) people that underwent a primary bariatric surgery operation in 2011-2013 were from an ethnic minority group. Only 24% of those who underwent bariatric surgery were male. Thus, the population that undergoes bariatric surgery in the UK is predominantly ‘White British’ and female. We aimed to over-represent men within our sample (41% of the post-operative participants in our study versus 24% nationally). As the numbers of people undergoing bariatric surgery from ethnic minority groups are very low, it is difficult to recruit them to qualitative studies that do not use large sample sizes, and this is a limitation (added to the Discussion, see response to later comment). Although our sample may not have been demographically diverse, a strength of our study is that participants were clinically diverse - the three main types of bariatric surgery procedures undertaken in the UK are represented, and participants were at various timepoints post-surgery (range of four months to nine years). They were also recruited from two different bariatric centres which had different follow-up care programmes and health professional teams, while the vast majority of qualitative studies in this area are single-centre.

It would be interesting to see some (brief) statistics of the kind of bariatric procedures that are in use in the UK as this may differ significantly between countries. For example, gastric bands are no longer in use in Sweden, due to the poor long-term results.

We have added information on bariatric surgery statistics in the UK compared with internationally into the 2nd paragraph of the Introduction.

“The three main types of bariatric operations performed in the UK include the Roux-en-Y gastric bypass (RYGB, 53.9% in 2011-13), the sleeve gastrectomy (SG, 21.4%), and the adjustable gastric band (AGB, 21.4%).¹⁴ More recent international data indicate that the SG (46.0%) and RYGB (38.2%) are the most common bariatric operations worldwide with AGB decreasing in recent years (5.0%), and the one-anastomosis gastric bypass now gaining popularity.¹⁵”

Additionally, as BMJ Open is not a journal focused on bariatric surgery or even obesity, I think its readership would appreciate a brief explanation of the nature and expected outcomes of the different bariatric procedures.

We have added information on the nature and outcomes of bariatric surgery into the 2nd and 3rd paragraphs of the Introduction.

“Each of these procedures works slightly differently; mechanisms include restriction in the amount of food able to be consumed, reduction in hunger, improvement in satiety, shift in food preferences, as well as altered gut hormones, bile acids, and vagal signalling.¹⁶ Whilst there are lots of non-randomised studies in this field, there are very few well designed and conducted randomised controlled trials with long-term follow-up. This means that true comparative assessments of RYGB, SG and AGB are absent from the literature. A current UK study has recently completed recruitment (n=1351), with the primary end point at three years. This will be the first pragmatic large-scale study examining all three procedures.¹⁷

Studies which have examined HRQL after each procedure are often poorly conducted with few including baseline data and comprehensive assessments of HRQL. Some show certain aspects of HRQL to improve but not others. 11 12 18...”

Methods:

In the Discussion section, you write that the main strength of the study lies in its “detailed qualitative approach”, but the Methods section gives very little information to justify this claim. The interviews are hardly described at all and I would like to see some more information about them – how long were they generally? If they were “mostly” conducted in the participants’ homes, where were the rest conducted, and how many were conducted outside the homes?

Additional detail about the interviews has been added into the 2nd paragraph of the Methods.

“Interviews were chosen as the method of data collection for this study due to the sensitive and complex nature of living with bariatric surgery, and to allow individual participants’ experiences to be explored in detail. Interviews were semi-structured to provide some consistency in topics discussed between interviews, while allowing flexibility to adapt each interview to the participant. Thirteen participants were interviewed in their homes, four in a private research room at one of the two participating hospitals, one in a private room at the University, and one over the telephone at their request. Interviews lasted between 44 and 110 minutes.”

Document S2 seems to contain a large number of questions irrelevant to the aim of the study under review, and it would be very helpful to have a short description of the questions you use for your results in the Methods section.

We have removed the first topic guide entitled ‘Pre-operative patient interviews’ in document S2 and included only the ‘Post-operative patient interviews’ topic guide as the findings reported in this paper are based on the interviews with these patients. We have also added a sentence into the 3rd paragraph of the Methods to specify which sections of the topic guide relate to the findings reported in this paper.

“Findings reported in this paper mainly relate to the sections of the topic guide ‘Actual outcomes of surgery’ and ‘Actual experiences of follow-up care’.”

In a similar vein, I would appreciate a more detailed description of the analytical process.

Additional detail on the analytic process has been added into paragraph 4 of the Methods.

“As the aim of the study was to broadly investigate patients’ experiences of surgery, including outcomes and aspects of care, this inductive approach to analysis was chosen to ensure that themes developed were strongly linked to the data. Coding was completed for all transcripts by KDC, with a sample of transcripts independently coded by two other experienced qualitative researchers (AOS and JLD) (see document S4 for final coding framework). Differences in interpretation were resolved through discussion. Initial codes were built into coding structures and themes were identified. Coding and data management were facilitated using NVivo 10 software.⁴⁰ Detailed descriptive accounts were written by KDC for each small batch of interviews, which described data relating to each theme and its constituent codes. It was at this stage that relationships between themes were identified, leading to the development of higher-order categories which encompassed inter-related themes. The coding and descriptive account were completed for each batch of interviews prior to recruiting additional patients so that emerging themes could be followed up to enrich subsequent interviews. Finally, large matrices were created to compare themes and categories across all participants and summary descriptive accounts were written wherein the concepts overarching all themes and categories crystallized.³⁹”

Results:

The results, though interesting, seem a little unfinished to me, like they would benefit from another round or two of consideration.

We have added two figures (Figures 1 and 2) showing the link between themes, categories and concepts which we hope improves clarity. We have also added some brief text to the first paragraph under ‘Adapting to life after surgery – normality and ambivalence’, and the 2nd paragraph under ‘Experiences of follow-up care – abandonment and isolation’ to make the link between categories and concepts a bit clearer.

“Throughout several areas of their lives, participants were striving to be more “normal” after bariatric surgery. This related to different aspects of their lives categorised as physical health, psychological health, eating patterns and hunger, body image, weight, and social functioning (Figure 1).”

“Overall, however, there was a sense of abandonment and isolation in participants’ accounts of follow-up care. This related to their experiences of post-operative support from the specialist team, primary care professionals, and peer support groups (Figure 2).”

In the first, introductory paragraph (page 7, lines 11-22) of the Results section, you mention “Adapting to life after surgery” and “Experiences of follow-up care” but not the middle category (“Social functioning and stigma”).

Also, since you use your two overarching concepts of “Normality” and “Ambivalence” as sub-headings in the other two categories, it seems natural to also have them in the category about social functioning and stigma?

We have now removed the sub-heading ‘Social functioning and stigma’ from the ‘Ambivalence’ section. This sub-heading was added to improve readability of the ‘Ambivalence’ section as there was a lot of data on social functioning and stigma that related to Ambivalence. We appreciate it may have caused confusion and so have removed it. We have also reduced the length of the text on social functioning to make the results tighter.

I also feel that the text under the different sub-headings isn't always entirely relevant to the that particular theme – for example, the section starting with “The majority...” (page 9, lines 5-8) doesn't seem to have anything to do with “Normality”.

We respectfully disagree with this comment. The paragraph referred to relates to the development of excess skin after bariatric surgery which participants reported to have a huge impact on their sense of normality. As described in the paragraph, although participants felt they were a more 'normal' size, without clothes on they felt much more abnormal due to their loose hanging skin.

I think the entire Results section could be re-structured so that it is tighter and the categories mutually exclusive, with all text under each category relevant to that particular category and no other.

We have edited and reduced text throughout the Results section to make the results tighter and more mutually exclusive. In particular we have reduced text under the previous 'Social functioning and stigma' sub-heading and have moved some of the text under 'Isolation' to 'Abandonment' to make the two sub-sections more mutually exclusive. Inevitably, however, there will be some data that are relevant to multiple themes or concepts (Nowell et al, 2017; Pope et al, 2006).

I notice that you refer to Normality and Ambivalence as “concepts” – would “themes” be a better word, considering your choice of analysis method?

We have used the terminology 'concepts' as described by Glaser & Strauss (1967) and Corbin & Strauss (2008) in their description of the constant comparison method which our analysis was based on. We have also added some brief text to the first paragraph under 'Adapting to life after surgery – normality and ambivalence', and the 2nd paragraph under 'Experiences of follow-up care – abandonment and isolation' to make the link between categories and concepts a bit clearer (as described earlier).

Discussion:

I note that you are quite generous with self-citation (four separate articles). There is nothing inherently wrong with citing one's own work, of course, but this may be a bit much. Could you replace at least one or two, do you think?

We have removed one of these references (reference 19 in the original version).

In exactly which way were the participants given “time and flexibility” (page 16, line 4)? Again, there is not enough detail in the Methods section to substantiate this claim.

Additional detail about the interviews has been added into the 2nd paragraph of the Methods, as described earlier.

I would also like to see a discussion on the relative homogeneity of the sample in the limitations section.

Additional sentences have been added into the 4th paragraph of the Discussion:

“A limitation of this study is the lack of ethnic diversity represented within the sample. Low numbers of people from ethnic minority groups undergo bariatric surgery in the UK (1303 between 2011-2013, 7.7% of total procedures), making it difficult to identify eligible people for qualitative studies.14 A

strength of this research is that we were able to over-represent male participants within our sample (41% of the 17 post-operative participants compared with 24% who undergo bariatric surgery nationally), which has been a limitation of previous qualitative studies in this area.^{14 28 30 32 34-36} An additional strength was the inclusion of a clinically diverse group of patients who had undergone all three main types of bariatric procedures in the UK and who were at a broad range of timepoints post-surgery. Participants were also recruited from two UK centres with different follow-up programmes and health professional teams.”

Reviewer 2 (Sandra Jumbe, Queen Mary University of London, United Kingdom):

Very well written paper and adds to much needed evidence on the psychosocial impact of bariatric surgery, particularly long term

Thank you for your support and comments. We have addressed each one individually below.

A few minor comments

- The introduction sets out the issues and study objective clearly. However, where you consider existing literature, there are a few other published qualitative papers similar to yours with patients who had several types of bariatric procedures that you could reflect on further.

Thank you for attaching your recent and very elegant study (Jumbe & Meyrick 2018) investigating post-bariatric surgery care, which unfortunately slipped through our net! We have added this study to the literature reflected upon in the Introduction (and Discussion) leading to the rationale for our study. We have edited the final paragraph in the Introduction to make the justification for the study stronger, including a clearer link with the objectives:

“The primary focus of most previous qualitative research in bariatric surgery has been on patient experiences of outcomes of surgery rather than experiences of follow-up care.^{10 19} Studies that have reported on aspects of care have identified patient need for longer follow-up after bariatric surgery, better access to psychological support, and the ability to communicate with health professionals between routine appointments.^{19 28-36} However, most of these studies were single-centre^{29-32 34 36} or reported findings from select groups, such as patients that had undergone one type of bariatric procedure only (e.g. adjustable gastric band)^{29 30 32-35} or had experienced negative outcomes such as weight re-gain or substance abuse issues.^{28 29 32 34} A recent systematic review by Parretti et al identified few studies focusing on patients’ experiences of follow-up care after bariatric surgery in the longer-term, and recommended that primary studies in this area were needed.¹⁹ The objectives of this study were to: 1) Investigate experiences of life after bariatric surgery including follow-up care in the long-term across people that had undergone all three main types of UK bariatric procedures, and 2) Use these findings to provide recommendations for follow-up care.”

- Very clear methodology. A bit more detail about why you chose interviews as a means of data collection and thematic analysis as an approach for data analysis in this context would be good

Thank you. We have added the following two sentences to the 2nd paragraph of the Methods about the choice of interviews:

“Interviews were chosen as the method of data collection for this study due to the sensitive and complex nature of living with bariatric surgery, and to allow individual participants’ experiences to be

explored in detail. Interviews were semi-structured to provide some consistency in topics discussed between interviews, while allowing flexibility to adapt each interview to the participant.”

The following sentence has been added into the 4th paragraph of the Methods relating to the analysis method:

“As the aim of the study was to broadly investigate patients’ experiences of surgery, including outcomes and aspects of care, this inductive approach to analysis was chosen to ensure that themes developed were strongly linked to the data.”

- Results - a lot of similarities with previous qualitative studies in terms of themes and subthemes which further strengthens evidence regarding life after surgery for those who have had bariatric surgery. Just one or two places where you could make your sentences shorter for better flow / easier read e.g. p10 lines 10-13

Thank you, we have reduced sentence length in the areas suggested, e.g.

“However, a number of participants had experienced a new type of social stigma at having taken the “easy way out” (P02) by having surgery (e.g. not achieved weight loss through the ‘normal’ means). Some were ashamed to tell others they had undergone surgery for fear of this reaction.” (page 11, line 23)

- Discussion

p14 line 2 remove 'in-depth' unless you are suggesting other qualitative studies are not in-depth?

We have removed 'in-depth' from this sentence to avoid any confusion.

p14 line 4 I am not sure 'represented' is the right word... perhaps illustrate, but just personal view

We have replaced the word 'represented' with 'illustrated'

p14 line 11 again I don't think 'ambivalence' is the right word here. I see that you are perhaps trying to link with the labels you have ascribed to your themes, but take a step back and think about it from a perspective away from this. I think this reflects the challenges or complexities of living with the surgical outcomes on a health perspective/ clinical level.

We have re-worded the sentence to: 'These complexities highlight the ambivalence of living with the outcomes of bariatric surgery.'

p14 line 22 - 25 some of the additional insights you state are not new. They have been found in previous qualitative studies therefore it would be good to reflect on these. A core text in this area is a book by Meana & Ricciardi (2008). I have attached another paper which has other qualitative studies in the reference list that you could review/consider in light of your findings

As mentioned earlier, thank you for bringing to our attention your recent and very relevant study (Jumbe & Meyrick 2018) investigating post-bariatric surgery care. We have added this study to the literature reflected up on in the Discussion (and Introduction as previously described). It is very encouraging to see parallels in findings between this study and ours. We have also included additional relevant references (including Meana & Ricciardi). The text in paragraph 2 of Discussion has been edited to read:

“Our findings are consistent with previous qualitative research on patient experiences of living with outcomes of bariatric surgery which depicted the complexities on patients’ sense of normality and the ‘give and take’ or ambivalent nature of the changes experienced.10 41-43 This study strengthens the evidence for the individual and nuanced nature of how bariatric surgery changes people’s relationship

with food in different ways, and changes over time, indicating the need for individualised dietary and psychological support at different time-points.^{10 28 41 43 44} The importance placed by participants on the social impact of bariatric surgery was also noted in a recent UK study by Graham et al.⁴⁵ These issues, including difficulties with social and family eating should be given more attention in follow-up care.”

The following text has also been added to paragraph 3 of the Discussion:

“Our study expanded the findings on patient experiences of bariatric surgery follow-up care as being characterised by feelings of abandonment and isolation, with views that services were not set up to support long-term issues. Abandonment was also evident in a study by Jumbe & Meyrick who described a “post-surgical cliff” with patients receiving intensive support prior to bariatric surgery and then feeling abandoned after surgery.³⁶ Similar to our study, they described how post-operative support was reliant on patient-initiated contact.”

We have also removed the sentences comparing our findings to the concept of recursivity from paragraph 3 to improve relevance and readability of this paragraph.

Are you aware of the new guidelines for psychological support that have just been published? Perhaps try to review/situate some of your discussion in context of these.

Thank you, I believe these were published just after this paper was submitted. We have added a sentence into the final paragraph of the Discussion.

“These recommendations are also in accordance with the recently published 2019 UK psychological guidelines for bariatric surgery which recommend a flexible and individualised approach to post-operative psychological support, including routine screening at 6-9 months post-surgery to identify support needs.⁵¹”

VERSION 2 – REVIEW

REVIEWER	Mikaela Willmer University of Gävle, Sweden
REVIEW RETURNED	02-Jan-2020

GENERAL COMMENTS	Thank you for your careful and thorough consideration of my comments and suggestions. I now feel that this manuscript is ready for publication.
---

REVIEWER	Sandra Jumbe Queen Mary University of London United Kingdom
REVIEW RETURNED	20-Dec-2019

GENERAL COMMENTS	Thank you authors for taking the time to adequately address previous reviewer comments. This has made for a much improved paper with a stronger rationale and updated references that link it well to current guidance/ pathways
--